# Psychological interventions for generalized anxiety disorder: Effects and predictors in a naturalistic outpatient setting

**Clara Krzikalla**, **Nexhmedin Morina**, **Tanja Andor**, **Laura Nohr**[¤], **Ulrike Buhlmann***

Institute of Psychology, University of Münster, Münster, Germany

¤ Current address: Department of Clinical Psychological Intervention, Free University Berlin, Berlin, Germany
* ulrike.buhlmann@uni-muenster.de

**Data Availability Statement:** The data cannot be shared publicly, because the informed consent and ethical approval for this study did not include that the anonymized data may be made freely available

## Abstract

### Objective

Numerous randomized controlled trials (RCTs) demonstrate the efficacy of cognitive behavioral therapy (CBT), metacognitive therapy (MCT), and methods to reduce intolerance of uncertainty (IU-CBT) in the treatment of generalized anxiety disorder (GAD). However, few studies have investigated these treatments under conditions of routine clinical care. The main objective of this study was to investigate the effectiveness of psychotherapy for GAD in an outpatient setting and to identify factors influencing treatment outcome.

### Methods

Fifty-nine GAD patients received naturalistic CBT (including MCT and IU-CBT) in an outpatient clinic and postgraduate training center for psychotherapy. Patients completed self-report questionnaires at the beginning and end of therapy regarding the main outcome worry as well as metacognitions, intolerance of uncertainty, depression, and general psychopathology.

### Results

Worry, negative metacognitions, intolerance of uncertainty, depression, and general psychopathology decreased significantly ($p's$ < .001) with large effect sizes for all symptoms ($d$ = 0.83–1.49). A reliable change in the main outcome worry was observed in 80% of patients, and recovery occurred in 23%. Higher worry scores at posttreatment were predicted by higher pretreatment scores, female sex, and less change in negative metacognitive beliefs during treatment.

### Conclusions

Naturalistic CBT for GAD appears to be effective in routine clinical care for worry as well as depressive symptoms, with particular benefits associated with altering negative metacognitions. However, a recovery rate of only 23% is lower than the rates reported in RCTs.

in public repositories. The data that support the findings of this study are available upon reasonable request from the psychotherapy outpatient unit of the University of Münster (contact via pta@uni-muenster.de).

**Funding:** The authors received no specific funding for this work.

**Competing interests:** The authors have declared that no competing interests exist.

Treatment needs to be improved, especially for patients with more severe GAD and for women.

## Introduction

Generalized anxiety disorder (GAD), which affects about 4% of the population at least once in their lifetime [1], is characterized by excessive anxiety and uncontrollable worry for at least six months, accompanied by physical or cognitive symptoms such as impaired concentration, irritability, or sleep disturbances [2]. GAD is associated with high rates of comorbidity [3, 4], impairment [5–7], long-term disability [8], and low quality of life [9]. Recent meta-analyses have reported medium to large pooled effect sizes for psychological interventions for GAD and medium effect sizes for comorbid depressive symptoms [10–12]. These meta-analyses further suggest that cognitive-behavioral therapy (CBT) is effective in treating GAD. CBT for GAD compromises applied relaxation, cognitive therapy, and exposure therapy, all of which have proven effective in randomized controlled trials [RCTs; e.g., 13–15; for an overview of CBT methods for GAD, see 16]. However, it is important to note that 40%–50% of patients undergoing treatment fail to achieve the necessary level of improvement on self-report questionnaires of worry to be considered recovered [17, 18].

Metacognitive therapy (MCT) was developed as a means of improving treatment efficacy [19]. This intervention is based on the metacognitive model of GAD, which highlights the importance of metacognitive beliefs about worry in the maintenance of the disorder. Positive metacognitions are beliefs about the need to engage in worrying, whereas negative metacognitions are beliefs about the uncontrollability and dangerousness of worrying [20]. Accordingly, MCT aims at modifying maladaptive metacognitive beliefs rather than the content of worry itself [21]. Several clinical trials indicate that MCT is an effective treatment over a range of disorders, including GAD [22].

Another rather new CBT approach is intolerance of uncertainty therapy (IU-CBT). Intolerance of uncertainty (IU) describes the tendency to react negatively to uncertain situations and events [23; for an overview of the construct, see 24]. IU is elevated in GAD patients and is suspected to lead to information processing biases [23]. IU-CBT includes worry awareness training, uncertainty recognition, behavioral and imaginal exposure, reevaluation of the usefulness of worry, and relapse prevention [25]. Several randomized controlled trials support the efficacy of IU-CBT for GAD [26–29].

Despite the evidence from RCTs, it remains unclear how effective psychological interventions for GAD are in routine clinical care. Specifically, patients with comorbid disorders are excluded in some clinical trials, which may limit the generalization of RCTs to routine clinical care [30, 31]. In addition, therapists in naturalistic settings might not use treatment protocols as developed in academic research [32] and, hence, produce different treatment effects. To address this, several studies have investigated treatment effects in naturalistic settings. In their meta-analysis, Hans and Hiller [33] reported large treatment effects for CBT in outpatients with anxiety and related disorders (but not GAD in particular) in routine clinical care. Stewart and Chambless [34] included in their meta-analysis 11 studies on generalized anxiety (in patients with GAD and other anxiety-related disorders) in naturalistic settings and found large pre- to posttreatment effects compared to benchmarking studies. Altogether, CBT seems to be effective in treating GAD in naturalistic settings. However, little information exists on the

administered methods and levels of psychopathology of study participants [35, 36], and strict treatment protocols have often been used [37–39], which limits generalizability.

Another relevant aspect concerns the factors influencing treatment efficacy. Meta-analyses have given inconclusive findings as to whether age and gender affect treatment outcomes [12, 40, 41]. Further, it remains unclear whether GAD patients with comorbid disorders benefit less from psychological interventions [41–43]. Only a few studies thus far have examined predictors of change in GAD directly in context with a therapeutic intervention [13, 44, 45] or in a naturalistic setting [44, 46–48]. Studies that examined predictors in routine care have either not focused exclusively on psychological interventions [46, 47] or have reported results across different anxiety disorders [44, 48] so predictors specific to GAD are not clear. With regard to metacognitions and IU, to our knowledge, there is no publication demonstrating how these factors are influenced by psychological interventions in routine care and whether this potential influence predicts treatment outcome.

Against this background, the current study aimed to elucidate the effectiveness of CBT (including methods from MCT and IU-CBT) for GAD in a naturalistic outpatient setting. We expected these treatments to lead to a significant reduction of worry as well as metacognitive beliefs, IU, depression, and general psychopathology from pre- to post-assessment. We further examined in an explorative way the role of age, sex, comorbidity, metacognitions, and IU on treatment outcome.

## Methods

### Participants

All participants were patients who presented on their own seeking treatment at the university's psychotherapy outpatient clinic between 2012 and 2020. The study included 59 patients ($M_{age}$ = 32.8 years, $SD_{age}$ = 11.5, range: 19–58 years, 66.1% women) with a primary diagnosis of GAD, as assessed with the Structured Clinical Interview for DSM-IV Axis I Disorders [SCID; 49]. Note that although the original DSM-5 was published in 2013 [2], the German version of the SCID based on DSM-5 was published only in 2019 [50]. For our study, we decided to interview all patients using DSM-IV-criteria. In total, 52.5% had at least one current comorbid disorder: The most common comorbid disorders were depressive disorders (28.8%) and other anxiety disorders (20.3%). For further information, see Table 1.

Eligibility criteria for inclusion into this study were (a) GAD as the primary diagnosis, (b) an indication for CBT, and (c) at least five sessions of psychotherapy. There were no specific exclusion criteria concerning any comorbidity or current medication status.

### Setting and procedure

The study took place at the outpatient clinic and postgraduate training center for psychotherapy at the University of Münster between 2012 and 2020. When contacting the clinic, patients received an initial appointment with licensed psychotherapists to check indication for outpatient therapy. Treatment was conducted in a naturalistic setting by psychologists in training to become psychological psychotherapists under close supervision by licensed psychotherapists. The postgraduate training included a specific workshop for treating GAD with CBT (including modern enhancements such as MCT and IU-CBT). Before therapy was requested via the health insurance provider, four to six diagnostic sessions occurred during which the pre-assessment took place. The post-assessment usually took place three to four sessions before end of therapy. Both the pre- and post-assessment consisted of self-report questionnaires filled out by the patient and basic documentation filled out by the therapist. Patients provided written informed consent to scientific use of the data at the beginning of the therapy either via the

**Table 1. Characteristics of patients at pre-assessment.**

| Baseline characteristic | *n* | % |
|---|---|---|
| Sex | | |
| Female | 39 | 66.1 |
| Age, *M* (*SD*) | 32.8 (11.5) | |
| Marital status | | |
| Single | 10 | 16.9 |
| Partnered | 28 | 47.5 |
| Married | 21 | 35.6 |
| Divorced | 0 | 0 |
| Housing[‡] | | |
| Alone | 12 | 20.3 |
| With partner/spouse | 34 | 57.6 |
| With children | 15 | 25.4 |
| With parents | 4 | 6.8 |
| Shared apartment | 10 | 16.9 |
| Highest educational level | | |
| Non-academic high school | 11 | 18.6 |
| Academic high school | 27 | 45.8 |
| University or postgraduate degree | 21 | 35.6 |
| Employment | | |
| Unemployed | 1 | 1.7 |
| In training | 27 | 45.8 |
| Employed | 27 | 45.8 |
| Self-employed | 4 | 6.8 |
| Native language | | |
| German | 54 | 91.5 |
| Other | 5 | 8.5 |
| Previous psychological treatment[†‡] | | |
| Outpatient setting | 22 | 37.3 |
| Inpatient setting | 12 | 20.3 |
| Psychotropic medication[‡] | | |
| Antidepressant | 11 | 18.6 |
| Neuroleptic | 3 | 5.1 |
| Gabapentinoid | 2 | 3.4 |
| None | 44 | 74.6 |
| Comorbid diagnoses[‡] | | |
| Depressive disorder | 17 | 28.8 |
| Other anxiety related disorder | 12 | 20.3 |
| Alcohol addiction | 1 | 1.7 |
| Somatization disorder | 1 | 1.7 |
| Adjustment disorder | 1 | 1.7 |
| Psychotic disorder | 1 | 1.7 |
| Psychological and behavioral factors associated with disorders or diseases classified elsewhere | 1 | 1.7 |
| None | 28 | 47.5 |
| Referral | | |
| By other healthcare provider | 22 | 37.3 |
| Directly | 28 | 47.5 |
| Counseling center | 2 | 3.4 |

(*Continued*)

**Table 1.** (Continued)

| Baseline characteristic | n | % |
|---|---|---|
| Other or unknown | 7 | 11.9 |

*N* = 59; no missing values appeared. Comorbid diagnoses according ICD-10 [51].

[†] Reflects the number and percentage of patients to whom this applies.

[‡] Multiple responses possible.

therapy contract or via a separate informed consent form. The procedure was approved by the Institutional Review Board of the Department of Psychology and Sport Science at the University of Münster.

Patients received an average of 51.4 sessions (*SD* = 18.6, range: 15–107). Treatment duration had a mean of 19.9 months (*SD* = 6.4, range: 6–45). Applied methods, number of sessions, and duration of treatment was not limited and varied according to patient needs. Therapists had the discretion, in consultation with their supervisors, to employ a combination of techniques from various approaches in the administration of treatment, and were not restricted to adhering to a specific treatment manual. A total of 82 patients consented to data analysis. Following treatment, the final treatment reports were rated by two independent evaluators (C.K. and a postgraduate psychologist in training) to ensure that GAD was the main focus of therapy and to determine the applied therapeutic interventions and comorbid disorders. We used a coding scheme to cluster interventions into the categories exposure therapy, cognitive restructuring, MCT, and IU-CBT. We excluded 23 patients because either GAD was not the main focus of therapy (n = 14) or patients took part in another manualized intervention study prior to therapy (n = 9). Interrater reliability was assessed with Cohen's kappa [52], κ = .78. Discrepancies between the two independent evaluators (n = 4) were resolved in joint discussion with a third clinically experienced person (T.A.). Following an intent-to-treat approach, the analyses were conducted with all patients who met the inclusion criteria. This included five patients who terminated therapy prematurely (8.5%). Most treatments combined different kinds of interventions: 79.7% of all patients received cognitive restructuring, 78.0% MCT, 76.3% exposure therapy, and 52.5% IU-CBT. In three cases (5.1%), only one therapy method was used. Additional methods included relaxation techniques, methods from acceptance and commitment therapy, emotion-focused therapy, and dialectic behavioral therapy.

## Measures

**Outcome measures.** To assess excessive worry as our primary treatment outcome, we applied the Penn State Worry Questionnaire [PSWQ; 53; German version: 54]. This self-report consists of 16 items rated on a 5-point Likert scale ranging from "not at all typical of me" to "very typical of me" (example item: "I worry all the time"). The PSWQ has good psychometric properties [55, 56], and the internal consistency in the present study was Cronbach's α = .83 for the pre-assessment.

The short form of the Metacognitions Questionnaire [MCQ; 57; German version: 58] includes 30 items measuring metacognitive beliefs about worry on a 4-point Likert scale ranging from "do not agree" to "agree very much". In this study, we used two of its five subscales: positive beliefs about worry (MCQ-POS, example item: "Worrying helps me to avoid problems in the future") and negative beliefs about the uncontrollability and danger of worrisome thoughts (MCQ-NEG, example item: "My worrying is dangerous for me"). The MCQ

possesses good psychometric properties [57–59]; in the current sample, the internal consistencies were α = .88 for MCQ-POS and α = .72 for MCQ-NEG.

To assess IU, we used the Intolerance of Uncertainty Scale [IUS; 60]. The shortened German version [61] consists of 18 items rated on a 5-point Likert scale ranging from "not at all characteristic of me" to "entirely characteristic of me" (example item: "Unforeseen events upset me greatly"). It has been shown to have good psychometric properties [61]; the internal consistency in the current sample was α = .93.

The Beck Depression Inventory-II [BDI-II; 62; German version: 63] is a self-report instrument consisting of 21 items, each with four statements with increasing severity, from which participants are asked to select the statement that best describes their condition over the past two weeks. A total sum of ≥9 indicates at least minor depressive symptoms. The BDI has high internal consistency, validity, and test-retest reliability [64, 65]; the internal consistency in the current sample was α = .88.

The Symptom Check-List [SCL-90-R; 66; German version: 67] measures general psychopathology as a self-report questionnaire with 90 items rated on a 5-point Likert scale. Participants are asked to rate how much they had been bothered by, e.g., headaches, crying easily, or overeating during the past week. The SCL-90-R has nine subscales and three summary scores; we used the summary score Global Severity Index (GSI, total sum for all items) as an indicator for general psychopathology for further analysis. The SCL-90-R has good psychometric properties [68], and the internal consistency in the current sample was α = .94.

**Potential predictors of treatment outcome.** Note that we also used the difference scores for MCQ-NEG, MCQ-POS, and IUS as predictor variables. We further examined the role of age, sex, and current comorbidity status.

## Data analyses

Statistical analyses were conducted with the statistical processing language R [69] mainly using the packages mice version 3.14.0 [70] and miceadds version 3.12–26 [71]. Prior to analyses, missing data were multiply imputed; 63.4% of all variables showed missing values. The amount of missing data ranged from 1.7% (PSWQ item 4 and several items on the SCL-90-R at pre-treatment) to 13.6% (MCQ item 10 at posttreatment). According to Little's MCAR Test [72], data were missing completely at random, $\chi^2 < .001$, $df = 4855$, $p = > .999$. We applied item-level and parcel summary multiple imputation using predictive mean matching [73, 74] as implemented in the R package mice [70] generating 50 sets of imputation. Auxiliary variables were identified using correlation analyses between the variables and missing variables as well as univariate $t$-test comparisons to improve the estimations and statistical power [75, 76]. We included all variables within the data set correlating ≥ ±.40 with one of the variables with missing values or with the missing variables (which indicate whether a data point is missing) as auxiliary variables into the imputation model. Further, we included variables which showed a significant univariate $t$-test and thus indicating statistically significant differences between participants with vs. without missing values. The potential scale reduction factor and graphical diagnostics were used for convergence diagnostics [75, 76]. Statistical analyses were performed separately on each completed dataset. Subsequently, all estimated parameters were pooled into a single set of results using Rubin's rules [77]. Pooling estimates from multivariate analysis of variance (MANOVA) is still the subject of research [e.g., 78]. Since there are currently no established pooling rules for repeated-measures MANOVA, we decided against pooling in this specific case and instead report the range of the 50 MANOVAs performed [79].

The assumptions for the statistical procedures [80] were examined in the original dataset. There were three univariate outliers in three different subscales based on $p < .001$ under the

z-distribution [81]. After inspection, it became clear that these were cases where either the initial symptom severity was low (pretreatment PSWQ) or the posttreatment values were higher than in the rest of the data (BDI and GSI). Removing or trimming these values might magnify effects, as this would only affect the patients showing a small pre-post difference. Therefore, we kept the values in the data. The violation of the multivariate normal distribution was responded to with an adequate statistical procedure (semi-parametric MANOVA). All other assumptions were met.

To analyze the outcome of the naturalistic treatment, we used a repeated-measures MANOVA, with time of measurement as the within-subjects factor and all outcome measures as dependent variables. Due to multivariate non-normality, we conducted a semi-parametric MANOVA using the multRM-function of the MANOVA.RM package version 0.5.3 [82]. This function outputs a Wald-type statistic (WTS), modified ANOVA-type statistic (MATS), and p values based on parametric bootstrapping (we applied 10,000 bootstrap resamples), which is recommended in a multivariate setting [83]. Post-hoc paired t-tests with Bonferroni-adjusted alpha levels ($\alpha_{adjusted}$ = .05/6 = .008) and effect sizes according to Cohen's formula, $d = (M1\text{-}M2)/SD_{pooled}$ [84], were calculated to determine effectiveness for each outcome variable. Due to the naturalistic study design, there is no control group; therefore, effect sizes might be inflated. To investigate clinical significance in addition to statistical significance, for the main outcome worry (PSWQ), we calculated reliable change and rate of recovery [85] based on criteria from Fisher [17]. In his review, he determined that a 7-point change on the PSWQ is needed for reliable change, and a cut-off for recovery of 46 points or below.

To inspect predictors of treatment outcome, we conducted a hierarchical multiple regression analysis with PSWQ posttreatment scores as the criterion. PSWQ pretreatment scores were entered in the first step to control for initial symptom severity. Age, sex, and comorbidity status were entered in a second step. In the final step, difference scores of MCQ-NEG, MCQ-POS, and IUS were entered in the regression model.

## Results

A table with the pooled correlations of all measures can be found in the supplementary material (S1 Table).

### Effectiveness of naturalistic psychotherapy

The repeated-measures MANOVA yielded significance in all 50 datasets (p values after parametric bootstrapping < .001 for WTS and MATS), indicating that the psychotherapy had an effect on the combined dependent variables. The WTS statistic (df = 6) ranged between 173.30 and 216.29, and MATS ranged between 269.25 and 302.26. Table 2 presents an overview of means, standard deviations, correlations, and effect sizes for the paired t-tests. All paired t-tests were significant apart from the pre- to posttreatment difference for MCQ-POS, which failed significance with the Bonferroni-adjusted alpha level of $\alpha_{adjusted}$ = .008. Effect sizes were highest for PSWQ (d = 1.49) and MCQ-NEG (d = 1.48).

Reliable change was achieved by n = 47 patients (79.66%, range: 46–48 in the 50 imputed datasets) with a reduction in PSWQ of at least 7 points. One patient deteriorated over the course of the treatment (1.7%). Considering recovery, only patients who were over the cut-off before the treatment were evaluated (N = 56). Of these, 13 patients (23.21%, range: 12–15) reached the cut-off criteria of 46 points or below, indicating recovery.

**Table 2. Pooled imputed values: Means (M), standard deviations (SD), t-values, p-values, and effect sizes (d) at pre- and posttreatment.**

| | Pre | | Post | | *t* | *p* | Cohen's *d* | 95% CI *d* |
|---|---|---|---|---|---|---|---|---|
| | *M* | *SD* | *M* | *SD* | | | | |
| PSWQ | 64.05 | 8.79 | 51.21 | 8.71 | 11.28 | **< .001** | 1.49 | [1.21, 1.88] |
| MCQ-NEG | 16.95 | 3.31 | 10.47 | 2.98 | 11.01 | **< .001** | 1.48 | [1.11, 1.86] |
| MCQ-POS | 10.63 | 3.84 | 9.40 | 2.61 | 2.43 | .019 | 0.33 | [0.06, 0.59] |
| IUS | 56.89 | 14.39 | 46.38 | 12.33 | 6.13 | **< .001** | 0.83 | [0.53, 1.13] |
| BDI | 17.54 | 8.82 | 7.62 | 7.56 | 7.80 | **< .001** | 1.02 | [0.71, 1.34] |
| GSI | 0.89 | 0.43 | 0.47 | 0.37 | 6.62 | **< .001** | 0.86 | [0.57, 1.17] |

Significant *p* values ($\alpha_{adjusted}$ = .008) are marked in bold. CI = confidence interval; PSWQ = Penn State Worry Questionnaire; MCQ-NEG = negative metacognitive beliefs of the Metacognitions Questionnaire; MCQ-POS = positive metacognitive beliefs of the Metacognitions Questionnaire; IUS = Intolerance of Uncertainty Scale; BDI = Beck Depression Inventory; GSI = Global Severity Index of the Symptom Check-List.

## Predictors of primary treatment outcome

The pooled results of the stepwise multiple regression for the PSWQ posttreatment score as the criterion are shown in Table 3.

Initial worry severity predicted higher posttreatment worry in all three regression equations. In the final model, PSWQ pretreatment scores explained 27% of the variance in posttreatment values. In the second step, only sex was shown to be an additional predictor and remained significant in the final model, explaining 7% of the variance. Based on the coding, it can be concluded that women have higher worry scores at posttreatment. Age and comorbidity status did not influence treatment outcome. Concerning MCQ-NEG, MCQ-POS, and IUS, only the difference score for MCQ-NEG was a significant predictor in the final model,

**Table 3. Pooled regression results using PSWQ as outcome measure.**

| Model | Predictor | *B* | *SE B* | *p* | β | *sr²* |
|---|---|---|---|---|---|---|
| a) | Intercept | 18.67 | 7.35 | **.01** | | |
| | PSWQ pre | 0.51 | 0.11 | **< .001** | .51 | .26 |
| b) | Intercept | 17.30 | 8.51 | **< .05** | | |
| | PSWQ pre | .48 | .12 | **< .001** | .49 | .22 |
| | Age | .01 | .09 | .91 | .01 | < .01 |
| | Sex[†] | 4.31 | 2.14 | **< .05** | .24 | .06 |
| | Comorbidity[‡] | -.51 | 2.00 | .80 | -.03 | < .01 |
| c) | Intercept | 20.26 | 7.56 | .01 | | |
| | PSWQ pre | .54 | .10 | **< .001** | .55 | .27 |
| | Age | -.01 | .08 | .92 | -.01 | < .01 |
| | Sex[†] | 4.87 | 1.89 | **.01** | .27 | .07 |
| | Comorbidity[‡] | -1.35 | 1.83 | .46 | -.08 | < .01 |
| | MCQ NEG Δ | -.88 | .24 | **< .001** | -.44 | .15 |
| | MCQ-POS Δ | -.05 | .27 | .86 | -.02 | < .01 |
| | IUS Δ | -.03 | .09 | .77 | -.03 | < .01 |

Significant *p* values (*p* < .05) are marked in bold. a) adjusted $R^2$ = .25 [.07, .45]; b) adjusted $R^2$ = .27 [.09, .47]; c) adjusted $R^2$ = .48 [.26, .66]; PSWQ = Penn State Worry Questionnaire; MCQ-NEG = negative metacognitive beliefs of the Metacognitions Questionnaire; MCQ-POS = positive metacognitive beliefs of the Metacognitions Questionnaire; IUS = Intolerance of Uncertainty Scale. Δ = change scores.

[†] 0 = male, 1 = female;

[‡] 0 = no comorbid disorders; 1 = at least one comorbid disorder

explaining 15% of the variance in posttreatment PSWQ scores. Higher difference scores in MCQ-NEG predicted lower posttreatment PSWQ scores. The final model explained 48% of the variance in worry posttreatment scores. A post-hoc power-analysis was performed using the R-package "pwr" [86]. The analysis indicated a power of .78 to detect the given effect, based on the adjusted $R^2$ = .26 (the lower bound of the 95% confidence interval for the $R^2$ for the final model) and the use of all seven predictor variables. A sample size of n = 85 would have been required to achieve a power of .95.

## Discussion

This study provides further insights into the treatment for GAD in routine clinical care by determining the effectiveness of CBT (including methods from MCT and IU-CBT) and identifying important predictors of treatment outcome in a naturalistic outpatient setting. Patients showed significant reductions in worry, negative metacognitions, IU, depression, and general psychopathology with large effect sizes from pre- to posttreatment. Eighty percent of the patients showed reliable improvements in worry, and 23% recovered. Only one patient deteriorated. Higher worry at posttreatment was predicted by higher symptom severity at pretreatment, female sex, and less change in negative metacognitions.

Naturalistic psychotherapy for GAD seems to be effective in not only reducing worry but also reducing general psychopathology and depressive symptoms. Effect sizes were large and consistent with previous literature [10, 33]. To our knowledge, this is the first publication to also investigate the influence of naturalistic psychotherapy on metacognitions and IU, which are known to play an important role in GAD [e.g., 24, 87]. Negative metacognitions were significantly reduced from pre- to posttreatment with a large effect size; yet, there was no significant effect on positive metacognitions after Bonferroni correction. This aligns with the metacognitive model, which specifically emphasizes the influence of negative metacognitions in the maintenance of GAD [87]. The large effect of naturalistic psychotherapy on IU is comparable to other studies using CBT approaches [88, 89]; this is particularly noteworthy, since strategies to reduce IU were only explicitly addressed in about 50% of therapies under study. However, it seems that IU might also be altered when not explicitly addressed [27].

Alongside statistical significance, we also investigated clinical significance. Eighty percent of the patients showed reliable change in terms of the main outcome worry. This is in line with or even exceeds previous studies under routine conditions [35–37]. However, only 23% of the patients recovered by the end of treatment. Although this is lower than the recovery rates of approximately 50% that have been found in RCTs [17, 18], our rate of 23% is within the range of other naturalistic studies, in which recovery rates ranged from 13% to 50% [36, 37, 43]. In RCTs and, as it seems, even more in naturalistic studies, there is still room for improvement in psychotherapy for GAD, as a big proportion of patients still fall into the clinical range at the end of treatment. One way to improve treatment is to better understand what factors influence the outcome of psychotherapy.

Therefore, the current study also examined which factors predicted worry at posttreatment. We found that pretreatment symptom severity predicted posttreatment symptom severity, such that patients with higher initial symptom severity remained more burdened with worry after treatment; this adds to a previously inconclusive body of studies [41]. In addition, after controlling for pretreatment symptom severity and other predictors, female sex predicted higher worry at posttreatment. Most previous studies have not found a sex-specific effect [12, 41], but those that have found this effect noted that it was in the same direction that we found in the current study [47, 90]. Since the current study included more female than male participants, this could also have influenced the rather poor recovery rate in the whole sample.

Overall, these results highlight the need to improve psychotherapy for GAD. Treatment in routine clinical care should especially be optimized for patients with more severe manifestations of GAD and women, as they continue to have worse scores at posttreatment. Age and comorbidity appear to require less consideration, as these factors failed to predict treatment outcome. Although some dissenting findings exist, our findings largely match previous evidence [12, 41, 46].

Concerning treatment factors, only change in negative metacognitions predicted worry at posttreatment. This fits with the theoretical framework of MCT and previous literature on the efficacy of MCT [22]. While IU decreased significantly over the course of treatment, this change did not predict treatment outcome in worry. This contradicts previous findings that have shown an association between changes in IU and changes in worry [24]; however, the studies cited by Carleton [24] did not control for metacognitions. The current study suggests that negative metacognitions play a more important role in reducing worry than IU. Notably, in the current study, IU was explicitly addressed in only about 50% of the treatments. More research is needed to further investigate the role of IU and metacognitions on treatment outcomes for GAD. To date, therapists in routine clinical care should focus on changing negative metacognitive beliefs to reduce worry.

## Strengths and limitations

The current study is representative of the target population and provides relevant insight into patient and treatment characteristics. Our sample consisted of patients with pretreatment worry scores comparable to such scores in RCTs [40] and in naturalistic settings [91]. It included patients with comorbidities and medication, as is often seen in routine clinical care. In addition, dropout was low (8.5%), and data was analyzed by applying an intent-to-treat-approach with an adequate method to deal with missing data, which is often lacking in other studies [10]. Treatment outcome was analyzed using multiple methods to gain insights into statistical and clinical significance, which was particularly important because discrepancies occurred between different methods. In this study, no treatment protocols restricted therapists to use and/or omit certain methods. In 95% of treatments in the current study, therapists mixed interventions from different treatment approaches (CBT, MCT, IU-CBT); such an eclectic approach is typical for therapists in routine clinical care [32, 92]. Nevertheless, this prevented us from making any statements about the efficacy of individual treatment methods.

In addition to the strengths, some important limitations must also be considered when interpreting the current study's results. Although the naturalistic design has advantages regarding external validity, internal validity might be compromised. First, because there was no control group, it is not possible to establish a causal relationship between the change in outcome measures and the psychotherapy administered. Further, the treatment duration averaged 20 months, leaving time for maturation, external factors, or other threats to internal validity. Due to the naturalistic design of effectiveness studies, they usually lack control groups. Nevertheless, it is important to study treatment effects in this setting, as the effects may differ from those in RCTs [30, 31]. Another limitation of this study is the reliance on self-report questionnaires as a method of data collection, which can be affected by bias [93]. However, a meta-analysis of GAD patients found that the effect sizes based on self-report questionnaires were smaller than those based on clinically-rated instruments [94], suggesting that the results based on self-report questionnaires may not overestimate the effectiveness of therapy. While the sample size provided sufficient power in this study, future replication efforts could benefit from a larger sample size to enhance the robustness of the findings and provide stronger

evidence for the conclusions drawn. Overall, results of the current study are comparable to other studies under routine conditions [e.g., 35, 37, 91].

## Conclusions

The current study demonstrated the effectiveness of CBT (including methods of MCT and IU-CBT) in reducing worry, negative metacognitions, IU, depressive symptoms, and general psychopathology in routine clinical care. Effect sizes were large and comparable to those of previous studies. It seems especially beneficial to focus on changing negative metacognitions, as this predicted a better outcome. Yet, in terms of clinical significance, only 23% of patients achieved recovery status. In particular, women and patients with more severe GAD at baseline had poorer treatment outcomes. Additional research is necessary to further increase the effectiveness of CBT with respect to the recovery rate of GAD in routine clinical care.

## Supporting information

**S1 Table. Pooled correlations between all measures.**
(DOCX)

## Acknowledgments

The authors would like to thank Katharina Schwieters for her assistance in the study.

## Author Contributions

**Conceptualization:** Clara Krzikalla, Nexhmedin Morina, Tanja Andor, Ulrike Buhlmann.

**Data curation:** Clara Krzikalla, Laura Nohr.

**Formal analysis:** Clara Krzikalla, Laura Nohr.

**Methodology:** Clara Krzikalla, Nexhmedin Morina, Tanja Andor, Ulrike Buhlmann.

**Supervision:** Nexhmedin Morina, Tanja Andor, Ulrike Buhlmann.

**Writing – original draft:** Clara Krzikalla.

**Writing – review & editing:** Nexhmedin Morina, Tanja Andor, Laura Nohr, Ulrike Buhlmann.

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
