## [Decision Letter · Decision Letter 0]

25 Jan 2023

PONE-D-22-24983Psychological interventions for generalized anxiety disorder: Effects and predictors in a naturalistic outpatient settingPLOS ONE

Dear Dr. Buhlmann,

Thank you for submitting your manuscript to PLOS ONE. After careful consideration, we feel that it has merit but does not fully meet PLOS ONE’s publication criteria as it currently stands. Therefore, we invite you to submit a revised version of the manuscript that addresses the points raised during the review process.

We look forward to receiving your revised manuscript.

Kind regards,

Stephan Doering, M.D.

Academic Editor

PLOS ONE

Journal Requirements:

Reviewers' comments:

Reviewer's Responses to Questions

**Comments to the Author**

1. Is the manuscript technically sound, and do the data support the conclusions?

Reviewer #1: Yes

Reviewer #2: Partly

2. Has the statistical analysis been performed appropriately and rigorously? 

Reviewer #1: I Don't Know

Reviewer #2: No

3. Have the authors made all data underlying the findings in their manuscript fully available?

Reviewer #1: No

Reviewer #2: No

4. Is the manuscript presented in an intelligible fashion and written in standard English?

Reviewer #1: Yes

Reviewer #2: Yes

5. Review Comments to the Author

Reviewer #1: This is a well-written paper that could surely contribute to the existing literature, however I have some suggestions to be addressed. First, some comment on the authors' methods is necessary as to why they chose to evaluate participants using the DSM IV as opposed to the DSM V (timeline etc.). Second, some self-report measures were employed in this study and other research shows us that these can sometimes be unreliable for various reasons. Should this be noted as a partial limitation of the study? Overall, this is a high quality manuscript fit for publication if these concerns are adequately addressed.

Reviewer #2: Abstract

- The authors may consider rewording “significant changes” and instead provide detail about the direction i.e., decreased.

- Could the authors please provide a brief explanation of how reliable change and recovery were calculated? Was this the RCI or a percentage change?

Introduction

- The introduction may benefit from combining the first two paragraphs together. Furthermore, could the authors please explain how recovery was conceptualized by the studies mentioned?

- The authors argue that previous studies of psychotherapy for GAD in routine care are limited by their strict treatment protocols. Did the current study have unstandardized treatment protocols? One could argue that psychology training clinics have rather strict protocols as the trainee psychologists are under close supervision and are often following treatment manuals.

- One strength of this paper is their investigation of IU and metacognitions as predictors of change in a naturalistic setting. However, the authors do note that previous studies have already examined predictors of change in GAD in naturalistic settings. The introduction could benefit from highlighting how the current study differs from these previous studies (refs 44-47).

Method

- The treatment duration of M = 51.4 sessions is long, particularly for exposure therapy or cognitive restructuring. Could the authors please provide the average session duration for each of the treatment approaches (perhaps the primary treatment type)?

- A sample of 59 participants is small for predictor analyses. Could the authors provide a power analysis or reference which justifies the use of n = 59 participants for these analyses?

- How did the authors conduct multiple imputation? What predictors were used to generate the replacement values?

- The authors may reconsider the use of repeated-measures MANOVA for treatment outcome and instead use mixed linear models or generalized estimating equations, as is standard in current psychotherapy research. For reference, see doi: 10.1001/archpsyc.61.3.310.

Discussion

- The authors state that this study provides insights into the effectiveness of CBT. They may reconsider replacing “CBT” with the umbrella term “psychotherapy” considering the diversity of treatment protocols included in the study.

6. PLOS authors have the option to publish the peer review history of their article (what does this mean?). If published, this will include your full peer review and any attached files.

Reviewer #1: No

Reviewer #2: No

---

## [Author Response · Author response to Decision Letter 0]

14 Feb 2023

Editor‘s comments:

Our response: We have revised our manuscript accordingly.

Our response: We have added a paragraph in the cover letter to clarify why we cannot publicly share the data for the study, even though we acknowledge and support the benefits of open data sharing.

Our response: It seems that we made this statement by mistake. As stated above, it will not be possible for us to share the data in a public repository. Our apologies for any confusion caused.

Our response: We have included captions for our Supporting Information File at the end of the manuscript.

Reviewers' comments:

5. Review Comments to the Author

Reviewer #1: This is a well-written paper that could surely contribute to the existing literature, however I have some suggestions to be addressed. First, some comment on the authors' methods is necessary as to why they chose to evaluate participants using the DSM IV as opposed to the DSM V (timeline etc.). Second, some self-report measures were employed in this study and other research shows us that these can sometimes be unreliable for various reasons. Should this be noted as a partial limitation of the study? Overall, this is a high quality manuscript fit for publication if these concerns are adequately addressed.

Our response: 

1. We thank the reviewer for their positive feedback. Given that the German version of the structured clinical interview for DSM (SCID) based on the DSM-5 was only published in 2019, we used the DSM-IV-based SCID for the entire study. In response to this comment we now clarify in the manuscript that we used the DSM-IV-based SCID for the entire study (page 6). 

2. We agree that self-report questionnaires can be prone to bias. We have added a section on this limitation in the discussion section (page 18).

Reviewer #2: Abstract

- The authors may consider rewording “significant changes” and instead provide detail about the direction i.e., decreased.

Our response: We have adjusted this part of the abstract to clarify the direction of the effects.

- Could the authors please provide a brief explanation of how reliable change and recovery were calculated? Was this the RCI or a percentage change?

Our response: Thank you for this inquiry. For the calculation of the number of participants with reliable change and recovery, we relied on data from Fisher (Fisher PL. The efficacy of psychological treatments for generalised anxiety disorder. In: Davey GCL, Wells A, editors. Worry and its psychological disorders. Theory, assessment, and treatment. West Sussex: Wiley & Sons; 2006. pp. 359–77). In his reanalysis of five RCTs with n = 223 participants, a RCI greater than ± 1.96 corresponds to a 7-point change on the PSWQ. In our study, we classified the participants with a change in PSWQ ≥ 7 as those with reliable change. Accordingly, we used the calculated cut off score of recovery from the same review and counted how many of our participants reached a PSWQ score of 46 or below at post-assessment. We have rephrased the method section to make it clearer that we did not calculate a reliable change index based on our data (page 12).

Introduction

- The introduction may benefit from combining the first two paragraphs together. Furthermore, could the authors please explain how recovery was conceptualized by the studies mentioned?

Our response: We have combined the paragraphs and rephrased the sentence to point out that recovery was based on the score in self-report questionnaires.

- The authors argue that previous studies of psychotherapy for GAD in routine care are limited by their strict treatment protocols. Did the current study have unstandardized treatment protocols? One could argue that psychology training clinics have rather strict protocols as the trainee psychologists are under close supervision and are often following treatment manuals.

Our response: Thank you for this comment. We have added a sentence to the manuscript (page 8) to clarify, that they were free in consultation with their supervisors, to combine methods from different treatment approaches. The final treatment reports yielded that they made use of this possibility. Therefore, we see our setting as rather unrestricted; especially in comparison with RCTs with often strict treatment protocols structuring each session. 

- One strength of this paper is their investigation of IU and metacognitions as predictors of change in a naturalistic setting. However, the authors do note that previous studies have already examined predictors of change in GAD in naturalistic settings. The introduction could benefit from highlighting how the current study differs from these previous studies (refs 44-47).

Our response: We have reworded the end of the paragraph on previous studies (refs 44-47) concerning predictors of change in GAD to highlight the gaps in the literature that our study contributes to fill.

Method

- The treatment duration of M = 51.4 sessions is long, particularly for exposure therapy or cognitive restructuring. Could the authors please provide the average session duration for each of the treatment approaches (perhaps the primary treatment type)?

Our response: We agree that an average of just over fifty therapy sessions is long and we would have liked to provide information on the differences between the different treatment approaches. Unfortunately, we only have three treatments in which only one of the approaches was used. From the final treatment reports, it was not possible to determine which was the primary treatment type. As is often the case in naturalistic therapies, the therapists used an eclectic approach and mixed different interventions. Therefore, unfortunately, we cannot report the duration of the treatment for each approach. 

- A sample of 59 participants is small for predictor analyses. Could the authors provide a power analysis or reference which justifies the use of n = 59 participants for these analyses?

Our response: We appreciate the prompt to the rather small sample size of n = 59 for predictor analysis in our study. Due to the inconclusive findings in the literature concerning predictors of treatment outcomes in GAD, we decided to run an exploratory regression analysis and did not compute an a priori power analysis. However, as suggested we have added a post-hoc power analysis to our manuscript (page 15) based on the final model of the regression analysis and also added confidence intervals for the R2 to reflect the uncertainty of our results. Based on our data, we calculated a power of .78 to detect the effect of the lower confidence interval for total R2. This is just below the .8 value that is normally considered acceptable power. We also added a sentence in our discussion section to highlight the importance of larger sample sizes in future studies to enhance statistical power and solidify the findings presented. 

- How did the authors conduct multiple imputation? What predictors were used to generate the replacement values?

Our response: Thank you for this valuable comment. We added some more specific information on the imputation model and the auxiliary variables which were used to impute the missing values. In general, we followed the instructions by Enders et al. which are already mentioned in the manuscript. We added the following sentences (page 11): We included all variables within the data set correlating ≥ ±.40 with one of the variables with missing values or with the missing variables (which indicate whether a data point is missing) as auxiliary variables into the imputation model. Further, we included variables which showed a significant univariate t-test and thus indicating statistically significant differences between participants with vs. without missing values.

- The authors may reconsider the use of repeated-measures MANOVA for treatment outcome and instead use mixed linear models or generalized estimating equations, as is standard in current psychotherapy research. For reference, see doi: 10.1001/archpsyc.61.3.310.

Our response: We recognize that mixed linear models or generalized estimating equations are commonly used in current psychotherapy research. However, we carefully considered the assumptions and limitations of different analysis methods in consultation with statistical support and determined that repeated-measures MANOVA was an appropriate choice for our study given the research questions, structure of the data, sample size, and ability to compare with previous studies. In addition, our analyses are clear and simple, making them accessible to a wide audience, including non-statisticians. This is crucial, because one goal is to provide practical recommendations for routine psychotherapy.

Discussion

- The authors state that this study provides insights into the effectiveness of CBT. They may reconsider replacing “CBT” with the umbrella term “psychotherapy” considering the diversity of treatment protocols included in the study.

Our response: We appreciate the reviewer's concern regarding the use of the term "CBT". We understand CBT as a comprehensive approach in psychotherapy, including for example the third-wave method of MCT. We have thoroughly reviewed the manuscript and taken steps to clarify our definition of CBT. Since we have not included other approaches, such as psychodynamic or systemic therapy, we prefer to avoid using the broader term "psychotherapy" to avoid any misunderstandings.

We also made a minor adjustment consisting of the inclusion of the category "divorced" in Table 1 on page 6 of the manuscript.

Finally, we would like to thank you for your efforts and attentive comments on our manuscript. We feel that our manuscript has improved greatly with your input. We hope that we have adequately addressed your concerns and that you may consider the manuscript for publication.

---

## [Editor Report · Decision Letter 1]

27 Feb 2023

Psychological interventions for generalized anxiety disorder: Effects and predictors in a naturalistic outpatient setting

PONE-D-22-24983R1

Dear Dr. Buhlmann,

We’re pleased to inform you that your manuscript has been judged scientifically suitable for publication and will be formally accepted for publication once it meets all outstanding technical requirements.

Kind regards,

Stephan Doering, M.D.

Academic Editor

PLOS ONE

---

## [Editor Report · Acceptance letter]

2 Mar 2023

PONE-D-22-24983R1 

Psychological interventions for generalized anxiety disorder: Effects and predictors in a naturalistic outpatient setting 

Dear Dr. Buhlmann:

I'm pleased to inform you that your manuscript has been deemed suitable for publication in PLOS ONE. Congratulations! Your manuscript is now with our production department. 

Kind regards, 

on behalf of

Professor Stephan Doering 

Academic Editor

PLOS ONE